# Overview of Embedded Rust Operating Systems and Frameworks

**DOI:** 10.3390/s24175818

**Published:** 2024-09-07

**Authors:** Thibaut Vandervelden, Ruben De Smet, Diana Deac, Kris Steenhaut, An Braeken

**Affiliations:** 1Department of Engineering Technology (INDI), Vrije Universiteit Brussel, Pleinlaan 2, 1050 Brussels, Belgium; thvdveld@vub.be (T.V.); an.braeken@vub.be (A.B.); 2Department of Electronics and Informatics (ETRO), Vrije Universiteit Brussel, Pleinlaan 2, 1050 Brussels, Belgium; rubedesm@vub.be (R.D.S.); or diana.deac@campus.utcluj.ro (D.D.); 3Communications Department, Technical University of Cluj-Napoca, Memorandumului 28, 400114 Cluj-Napoca, Romania

**Keywords:** rust, embedded operating systems, performance evaluation

## Abstract

Embedded Operating Systems (OSs) are often developed in the C programming language. Developers justify this choice by the performance that can be achieved, the low memory footprint, and the ease of mapping hardware to software, as well as the strong adoption by industry of this programming language. The downside is that C is prone to security vulnerabilities unknowingly introduced by the software developer. Examples of such vulnerabilities are use-after-free, and buffer overflows. Like C, Rust is a compiled programming language that guarantees memory safety at compile time by adhering to a set of rules. There already exist a few OSs and frameworks that are entirely written in Rust, targeting sensor nodes. In this work, we give an overview of these OSs and frameworks and compare them on the basis of the features they provide, such as application isolation, scheduling, inter-process communication, and networking. Furthermore, we compare the OSs on the basis of the performance they provide, such as cycles and memory usage.

## 1. Introduction

An embedded Operating System (OS) is a software layer that particularly manages scheduling of tasks and hardware resources of an embedded device. For many years, these OSs have been developed in the C programming language. The C language is a compiled language that provides low-level access to memory, making it suitable for embedded devices. Unfortunately, it does not prevent the programmer from writing unsafe operations, such as indexing an array out of its bounds. This can lead to security vulnerabilities, which is a big concern, especially with the increasing number of interconnected devices.

The goal of Google’s “Project Zero” [1], which started in 2014, is to find security vulnerabilities in popular software. Each year, they publish a report with detected and reported zero-day vulnerabilities. They also keep track of these vulnerabilities in a publicly available dataset [2]. At the time of writing, this dataset contains 310 vulnerabilities, with 225 (72%) of them being caused by memory corruption, use-after-free, or type confusion. In a presentation at the BlueHat IL conference in 2019, Mat Miller, a security engineer at Microsoft said that the root cause of the vulnerabilities found in Windows products are from memory safety issues [3].

Tools that check for memory vulnerabilities do exist. However, software developers must know how to use them correctly and understand the reports that these tools generate. Designing a programming language that enforces safe memory manipulation would be a better solution.

In recent years, new programming languages suitable for embedded devices have emerged. The most prominent is the Rust programming language. It is a compiled language, providing safe low-level access to memory. The language has a strong type system and a set of rules that prevent unsafe memory operations. The compiler checks at compile time if the code follows these rules, and if not, the code will not compile.

Many papers discuss the performance of Rust programming language [4,5,6,7,8] and its safety  [9,10,11,12,13]. The conclusion of these works is that Rust can be considered a good replacement for C, yielding comparable performance while providing safety.

Ayers et al. [14], Uzlu and Şaykol [15], and Noseda et al. [16] focus specifically on embedded systems programming in Rust. They show that the design concepts of Rust are a good fit for embedded systems. With the ownership model and borrowing rules, peripherals can only be accessed by one task at a time. This prevents misuse of the peripherals, and is checked at compile time. The zero-cost abstractions allow for high-level Application Programming Interfaces (APIs) to be written, without having to worry about the performance of the code. However, these papers also show that compared to C, Rust binaries are larger, which can be a problem for embedded devices. They pinpoint that the increased binaries’ size is caused by generic code and by string formatting.

The structure of this work is as follows: in Section 2, we have a look at the recent literature on the Rust programming language, and in Section 3, we explain how the language is used in the embedded Rust ecosystem. We highlight core features of OSs. In Section 4, we discuss OSs and frameworks that are written in Rust and evaluate them in Section 5. Section 6 concludes the work.

## 2. Related Work

An important reference concerning the review and comparison of OSs for embedded devices is the work of Javed et al. [17]. The authors thoroughly examine eight OSs, designed specifically for Internet of Things (IoT) devices, written in C or C++. Their analysis covers a wide range of critical aspects, such as the type of kernel used by each OS, the scheduling mechanisms implemented, Inter-Process Communication (IPC) mechanisms, and the programming models and languages supported, as well as the network stack functionalities. Furthermore, the authors discuss memory management, energy efficiency, simulation tools, and application support provided by these OSs. We consider the same core features for our evaluation of Rust OSs for embedded systems.

Another work that discusses core features of OSs for embedded devices is  [18]. In this work, the authors discuss the main features of eleven OSs. This discussion is given in the form of a table. However, the authors do not enter into detail beyond this table comparison.

The work of [19] provides a performance evaluation of three popular OSs for IoT devices. In their study, they benchmark Contiki-NG, RIOT-OS, and Zephyr, all written in C. They conduct a total of eight benchmarks, based on the Thread-Metric Benchmark Suite. These benchmarks evaluate OSs in various aspects, including context switching, interrupt processing, IPC, and memory management. They also analyze the memory footprint and energy consumption of each OS. We adopt a similar approach to evaluate the performance of Rust-based OSs. We consider interrupt processing, scheduling latency, and the memory footprint.

Of course, many other papers discuss and evaluate OSs for embedded devices. However, our work is the first to discuss and evaluate OSs entirely written in Rust.

## 3. Preliminaries

The first stable version of the Rust programming language [20] was released on 15 May 2015. One of the selling points of Rust is its compiler-guaranteed memory safety, enabled by its novel ownership model. This ownership model is a set of rules that defines what can and cannot be performed with memory. The compiler will check if the code follows these rules.

Rust has three ownership rules. (1) Each value in Rust has only one owner, e.g., a variable being the owner of stack memory. (2) There can only be one owner at a time. Two variables cannot share memory. (3) When an owner goes out of scope, the value is dropped, e.g., the memory is not in use any more and resources are freed. A variable’s memory ownership can be transferred to another variable. This mechanism, called *moving* of ownership, restricts the original owner from doing anything with the memory. The *borrowing* rules address the inconvenience of moving ownership all the time. At any given time, (1) there can be any number of immutable references to a variable, or (2) there can only be one mutable reference. Whenever ownership of a value is borrowed, the owner cannot do anything with the memory until the borrow is released.

With the *ownership* and *borrowing* rules, many memory safety problems are addressed, such as null pointer dereferencing, dangling pointers, use-after-free, and data races. The rules are checked at compile time, which minimizes runtime performance overhead. Some checks, such as array bounds checking, are performed at runtime. However, the compiler can often optimize these away if it can prove that they are not necessary.

The Rust ecosystem for embedded devices, asynchronous Rust, and the core features of OS and frameworks are discussed in Section 3.3.

### 3.1. Embedded Rust Ecosystem

Rust comes with three built-in libraries. The *core* library provides fundamental types and functions that are available in all Rust programs, without relying on the standard library (*std*). The *alloc* library extends the functionality of the *core* library by adding features related to memory allocation, deallocation, and data structures such as dynamically sized arrays. The *alloc* library is only available on platforms that provide a memory allocator. This is usually the case on desktop platforms, but not necessarily on embedded platforms. Although memory allocators can be implemented for these platforms, they often entail reduced runtime performance. Finally, there is the standard library (*std*) with a wide range of functionality that makes it easier to work with various data types, perform I/O operations, and utilize higher-level abstractions. The standard library relies on the presence of an operating system, and is usually not available in resource-constrained platforms.

For embedded devices, the program entry point needs to be defined. This is the position in code where the program starts executing. The program starts with initializing static memory and vector tables, and sets up the stack pointer. This code needs to be explicitly written by the programmer, unless libraries are available that take care of this job. On ARM Cortex-M platforms, the  cortex-m-rt [21] library can be used. It takes care of the aforementioned tasks and provides a macro to define the entry point. Libraries for MIPS-, RISC-V-, or ESP32-C-based microcontrollers also exist, being mips-rt [22], riscv-rt [23], and esp32c-rt [24], respectively. For Espressif-based microcontrollers, such as the ESP32 series, a fork of the Rust compiler is provided by Espressif Systems [25]. This modified Rust compiler is able to compile a modified standard library for these devices. This allows developers to write very-high-level code, without the trouble of handling the hardware. One downside is that it creates an Espressif-specific ecosystem, often incompatible with the rest of the embedded Rust ecosystem.

After the setup performed by cortex-m-rt or mips-rt, the microcontroller peripherals need to be initialized, such as UART, I2C, and SPI. This is conducted by accessing the registers of the microcontroller. The description of the registers available in a microcontroller is usually provided by the manufacturer in the form of an System View Description (SVD) file. Using the svd2rust [26] tool, a Rust Peripheral Access Crate (PAC) is created using the SVD file. A Rust “crate” is what a library is called in Rust. This PAC contains a type safe API to access the peripherals of the microcontroller. svd2rust supports the following architectures: ARM Cortex-M, MSP430, RISC-V, and Xtensa LX6.

Now that peripheral registers can be accessed using a type safe API, Hardware Abstraction Layers (HALs) can be used to abstract more parts of peripherals. The Rust Embedded Working Group has a set of common and recommended guidelines for writing HALs. One of their guidelines is that types in HALs should implement the traits defined in the embedded-hal crate. Traits allow for defining shared behavior without specifying how the behavior is implemented, much like interfaces in object-oriented programming. The embedded-hal crate contains traits for types that abstract over embedded hardware. This crate offers an abstraction for building a platform-agnostic embedded ecosystem. Other crates provide traits for types that abstract over embedded hardware. Examples are embedded-nal and embedded-storage. The former contains traits for types that abstract over embedded networking, where the latter contains traits for types that abstract over storage such as EEPROM, NOR-flash, etc. Using these traits, HAL developers can enable the use of existing drivers for their platform for devices, such as sensors and actuators. For driver developers, the use of traits can make the driver generic and, thus, available for any platform that makes use of the embedded-hal crate. Although these abstraction layers seem to add a lot of overhead, it is often optimized away by the Rust compiler.

### 3.2. Asynchronous Rust

Asynchronous programming is a programming model where tasks are independently waiting for some resource to become available without blocking each other. An example is a task that makes use of hardware communication peripherals, e.g., I2C, SPI, or UART. Waiting for these peripherals to become available might block the microcontroller, wasting battery and time.

The Rust asynchronous model is built around the Future trait, representing an asynchronous computation. A *future* waiting for a resource will not block, but instead return and be scheduled to wake up again later. A future needs to be *polled* for it to be able to complete its computation. This is performed by calling the poll function on a future, which returns a Poll enum. The Poll enum has two possible values, either Pending or Ready(T), with T being the value the future returns when ready. Continuously calling the poll function until it is ready is obviously not optimal. As an argument, the function receives a Context, with which it is able to notify a scheduler, called an Executor in Rust, when it should be polled again.

Figure 1 depicts the asynchronous model in Rust, showing an executor and two futures. The executor polls the first future, which is executed until it is ready, or until it needs to wait for a resource. To indicate that it is waiting, the future returns Poll::Pending to the executor. The executor then polls the second future, which is executed in the same way. An interrupt routine notifies the executor that the first future is ready to be polled, which makes the executor poll the first future again. The first future will then be ready and will return Poll::Ready to the executor, telling it not to poll that future anymore.

The asynchronous model is embedded in the Rust language which facilitates the writing of asynchronous code. With the async keyword, a function can be defined as asynchronous. This function returns a future, which can be awaited by calling .await on the future. Note that futures are lazy, meaning that they are not executed until they are polled. Scheduling and polling of futures is performed by an executor, which is a runtime that manages these futures. Without an executor, futures cannot be executed.

### 3.3. Operating Systems and Frameworks

Using the HALs and PACs described in Section 3.1, one can develop an OS or framework. The primary function of an OS is to handle low-level tasks. In contrast, a framework typically offers a collection of libraries and tools designed to facilitate application development. An OS can be built using a framework as its foundation. The following sections go deeper into the core features of an OS: task scheduling, application isolation and memory protection, IPC, hardware interaction, and networking.

#### 3.3.1. Task Scheduling

Task scheduling is an important part of an OS or framework. An OS usually runs multiple applications on a microcontroller, where each application is a separate task or a set of tasks. All these tasks need to be scheduled in such a way that each of them obtains its fair share of microcontroller time.

Tasks can be scheduled *preemptively* or *non-preemptively*. With preemptive scheduling, a scheduler can interrupt and stop the execution of a running task, such that the new task can be executed with the intention to continue the interrupted task later. In non-preemptive scheduling, a task runs until it completes or voluntarily yields control to allow other tasks to execute. Tasks that voluntarily yield control are called *cooperative* tasks. Whereas non-preemptive scheduling is simpler to implement, it is usually not suitable for real-time systems, as it cannot guarantee strict deadlines. The following scheduling algorithms are commonly used in embedded systems:**Priority-Base** Each task is assigned a priority, and the task with the highest priority is scheduled first. Priorities are usually assigned by the developer.**Round-Robin** Each task receives a time slice, after which the next task is scheduled. Tasks are always scheduled in the same order.**Rate-Monotonic Scheduling** This is a priority-based scheduling algorithm where the priority of a task is determined by its period. The period of a task is defined by the time it takes until the task needs to yield control again. Tasks with shorter periods have higher priorities.**Earliest Deadline First** This is a priority-based scheduling algorithm where the priority of a task is based on its deadline. Tasks with the earliest deadlines have the highest priority.**Multi-Level Feedback Queue (MLFQ)** Tasks are grouped into different queues, e.g., based on their priority. Each queue has its scheduling algorithm, which is tailored to the tasks in that queue.

These scheduling algorithms can be combined, allowing a task to have both a priority and a deadline. In such cases, the task with the highest priority and the earliest deadline is scheduled first. There are many more scheduling algorithms, each with its use cases and advantages, but the ones mentioned above are the most common in embedded systems.

#### 3.3.2. Application Isolation and Memory Protection

Application isolation and memory protection are essential security features in an OS, particularly in the IoT domain. They ensure that tasks cannot access memory outside their designated permissions for reading or writing. Isolation is typically achieved through virtual memory, where each task operates within its own protected virtual address space, mapped to physical memory by the OS. Hardware mechanisms, such as the Memory Protection Unit (MPU), enforce strict access controls by defining specific memory regions accessible to each task. If a task attempts to access unauthorized memory, the MPU triggers a fault to prevent the operation.

#### 3.3.3. Inter-Process Communication (IPC)

IPC is the part of the OS that defines how tasks share data with each other. This is especially relevant when application isolation or memory protection is present. There are many ways to implement IPC, each with their use cases and advantages. Implementing IPC efficiently is crucial for the performance of the OS, providing responsive and reliable applications. This can be achieved in different ways:**Shared Memory** Tasks can access and modify the same memory region, which can be managed safely through synchronization mechanisms like semaphores or mutexes.**Message Passing** Tasks communicate with each other using message queues, mailboxes, or similar message passing systems, providing a structured approach to sharing information between tasks. This method ensures that synchronization is handled by the message passing system, reducing the likelihood of errors compared to shared memory.**Signals** Signals are a special form of message passing, where the message is very simple. This mechanism enables efficient event notification between tasks.

Each method has its own advantages and is chosen based on factors such as the type of data being communicated, synchronization requirements, latency considerations, and system architecture.

#### 3.3.4. Hardware Interaction

Embedded applications often require interaction with hardware peripherals such as sensors and actuators, which are controlled via registers. To facilitate the interaction with these registers, the OS or framework must offer an API that ensures type safety, ensuring that only appropriate registers are accessed, as verified by the compiler. As discussed in Section 3.1, the svd2rust tool can generate a PAC from an SVD file. This PAC provides a type-safe API for accessing the microcontroller’s registers. HALs can then be developed on top of these PACs, abstracting more complex aspects of the peripherals. Some OSs and frameworks provide custom HALs tailored to their specific abstractions and requirements. This is often mandated by their IPC, application isolation, and memory protection design choices.

#### 3.3.5. Networking

The Rust core library does not provide networking support, so network stacks must be implemented by the OS or framework. It is an essential component in the IoT domain, allowing devices to communicate with each other over wired or wireless networks. Most network stacks are implemented in the OS as part of the kernel, or in a separate library. Platforms that support the Rust standard library can utilize its networking stack.

For wired connections, network stacks typically support Internet Protocol version 4 (IPv4), whereas for wireless IoT connectivity, Internet Protocol version 6 (IPv6) is more common. However, IPv6 packets are often too large for low-power wireless networks, so the IPv6 over Low-power Wireless Personal Area Networks (6LoWPAN) protocol is used to run IPv6 over these networks. Common transport layer protocols are User Datagram Protocol (UDP) and Transmission Control Protocol (TCP), with UDP being more common in the IoT domain due to its lower resource requirements. Constrained Application Protocol (CoAP), Message Queuing Telemetry Transport (MQTT), and Matter are popular application layer protocols in the IoT domain.

The most important network stack in the Rust ecosystem is smoltcp [27]. It is a standalone network stack that targets bare-metal systems and supports common protocols like IPv4, IPv6, 6LoWPAN, UDP, and TCP. It can also process Ethernet and IEEE 802.15.4 frames. Smoltcp provides a Device trait, which defines how the network stack can interact with the hardware. Adding support for a device requires implementing this trait for the specific hardware. The smoltcp interface is polling-based, requiring the user to call the poll function on the network stack to process incoming and outgoing packets. This polling-based interface maps well to the asynchronous model in Rust.

## 4. Rust-Based Operating Systems and Frameworks

Since the start of the Rust project, many OSs and frameworks targeting embedded devices have been developed. In this section, we discuss features like scheduling mechanism, IPC, hardware interaction, and networking capabilities for some of these. We will evaluate four OSs and frameworks that are well known and used in the Rust ecosystem in detail.

### 4.1. Overview

For the selected OSs and frameworks, Table 1 shows the targeted platforms, licensing, number of contributors, git commits, lines of code, and GitHub stars. All four support ARM Cortex-M-based microcontrollers. We observe that some of them also start to support RISC-V-based microcontrollers. We also noticed that Tock [28] is the most popular OS in the Rust ecosystem, followed by Embassy [29] and RTIC [30]. Despite Hubris [31] being less popular, it still has an interesting approach for building a secure and robust OS.

#### 4.1.1. Tock

Tock [32] is an OS targeting Cortex-M- and RISC-V-based embedded platforms. Its development started around 2015, with the start of Rust, and is available under Apache 2.0 or MIT license, at the user’s choice. Licensing like this is common in the Rust ecosystem. Tock’s primary objective is to execute multiple untrusted applications, with its emphasis on shielding against potentially malicious applications and device drivers. The Tock developers have written their own network stack, which has support for 6LoWPAN and Thread over IEEE 802.15.4 links.

#### 4.1.2. Hubris

Hubris [31], developed by the Oxide Computer Company, is an OS that only targets 32-bit ARM Cortex-M microcontrollers. Although Hubris targets ARM Cortex-M microcontrollers, they have a limited set of supported boards. This is because they only support devices that they use themselves. Hubris is licensed under the Mozilla Public License Version 2.0. One of the goals of Hubris is to provide a robust and secure OS for embedded devices. They achieve this by using a strict task model, where each task is defined at compile time.

#### 4.1.3. RTIC

RTIC [30], short for Real-Time Interrupt-driven Concurrency, is a hardware-accelerated Real-Time Operating System (RTOS) based on research from the Luleå University of Technology (LTU) Sweden. It is licensed under the Apache 2.0 license or the MIT license, at the user’s choice. It targets ARM Cortex-M microcontrollers, with support for RISC-V in experimental stage. RTIC only provides an execution framework, meaning that it only provides the tools to write the execution and scheduling of tasks. HALs from other sources, such as the ones from the Rust ecosystem, should be used for interfacing with the hardware.

#### 4.1.4. Embassy

Embassy [29], originally developed by Dario Nieuwenhuis, is a framework that provides an asynchronous runtime (see Section 3.2) for embedded devices. It is licensed under the Apache 2.0 license or the MIT license, at the user’s choice. Embassy’s asynchronous runtime has support for ARM Cortex-M, RISC-V, and AVR platforms. It can also be used on platforms that support the std library, such as Linux or Windows. Furthermore, it can also be used on a WebAssembly architecture, although its primary focus is on embedded devices.

#### 4.1.5. Other Notable OSs and Frameworks

There are many more OSs and frameworks that are written in Rust. However, these are either not as well known or not as mature as the ones discussed above, or their development has halted. The following list does not aim to be exhaustive:**Aerugo (2023–2024)** [33] was developed as part of the European Space Agency activity to evaluate Rust for space applications. It is a real-time OS for ARM Cortex-M microcontrollers, more specifically, the ATSAMV71Q21 microcontroller.**Bern RTOS (2021–2023)** [34] is a real-time OS for ARM Cortex-M microcontrollers. It was developed by Stefan Lüthi during his master’s thesis at the Bern University of Applied Sciences.**Drone OS (2017–2023)** [35] Valentine Valyaeff started with the development of Drone OS in late 2017; targeting ARM Cortex-M and RISC-V microcontrollers. It was halted at the beginning of 2023.**lilos (2019–ongoing)** [36] is a minimal async executor for embedded devices, even though it is presented as a real-time OS. It currently only supports the ARM Cortex-M architecture.**RIOT-rs (2024–ongoing)** [37] is developed by the RIOT community. It is a port of the RIOT OS to Rust. It is built on top of the hardware abstractions and async programming framework provided by Embassy. It provides new abstractions such as threads, which are not available in Embassy. While RIOT is a mature OS for embedded devices, RIOT-rs is still in the early stages of development, and is, therefore, not included in our evaluation.**Wasefire (2023–ongoing)** [38] is a research project of Google. Its goal is to provide a platform for embedded devices on which WebAssembly applications can run.

### 4.2. Discussion of Core Features

The following sections provide a detailed discussion on the core features of the four OSs and frameworks. Per key aspect, we provide the design decisions and architecture of each of the discussed projects.

#### 4.2.1. Task Scheduling

**Tock**: Tasks that can be scheduled by the Tock kernel are defined in kernel- and user-space. In kernel-space, these tasks are called capsules. A capsule provides a specific service, such as a driver for a hardware peripheral. In user-space, these tasks are defined by the user as applications. Multiple applications can be loaded onto the microcontroller, each running in their own tasks. Tock defines the Scheduler trait, such that the kernel can be generic over different schedulers. Tock provides multiple different implementations, currently either round-robin, priority-based, cooperative, or MLFQ.

The scheduler is used in the main loop of the kernel. When an interrupt occurs, the currently running task is preempted by the kernel task. It is then up to the scheduler to decide which task to run next. Kernel capsules and interrupts are scheduled first. These are scheduled cooperatively, meaning that they need to yield control to allow other tasks to run. The scheduler trait allows deferring handling of interrupts, but none of the current schedulers make use of this feature. User-space tasks cannot preempt other user-space tasks.

If all tasks are ready or waiting for a resource, the kernel will enter a sleep mode. Depending on the platform, this mode enters a low-power state, where the microcontroller consumes less power. When an interrupt occurs, the kernel wakes up and the scheduler is run to determine the next task to run.

**Hubris**: Hubris focuses on robustness and security. To achieve this, tasks are defined at compile time in an application configuration file. This means that tasks are known at compile time, as it is not possible to create tasks at runtime. This reduces the risk of failure due to memory exhaustion when creating tasks.

Hubris uses a priority-based scheduler, where each task is assigned a priority. The task with the highest priority is scheduled first. Lower priority tasks are scheduled when no higher priority tasks are ready to run. Tasks with the same priority are scheduled cooperatively. When an interrupt occurs, the currently running task is preempted, but only when the preempting task has a higher priority.

Hubris has a *supervisor* task, which has the highest priority. It must take action when other tasks fail. When a task fails, e.g., because of an explicit crash, a hardware fault, or system call fault, the supervisor task is scheduled to run. The supervisor task asks the kernel to restart the failed task. If the supervisor task should fail, the full system is restarted.

Another important task is the *idle* task, which has the lowest priority. This task is scheduled when no other task needs to run. It will place the microcontroller in a low-power state, waiting for an interrupt.

**RTIC**: RTIC provides two kind of tasks: hardware tasks and software tasks. Hardware tasks are bound to a hardware interrupt and are executed as a reaction to a hardware event. These tasks, though not enforced, should always run to completion as they handle real hardware interrupts. Software tasks are not bound to any hardware event, but bound to a *dispatcher*. This dispatcher is acting as an async executor. This executor is driven by an unused hardware interrupt. For each unused hardware interrupt, an executor is assigned to it with a priority level. The list of executors should cover the list of priority levels defined by the application.

RTIC uses preemptive priority-based scheduling, optimized using a Stack Resource Policy (SRP) [39] based concurrency and resource management model. This model prevents deadlocks when sharing resources between tasks. Each task has a priority level and defines what resources it is sharing with other tasks. Using this information, the highest possible priority level is calculated for each resource. A running task raises the preemption level to the highest priority level of all resources it is using. This way, a task can only be preempted by a task with a higher priority level. Tasks sharing resources can, thus, not be preempted by each other. This analysis is performed at compile time, where the tasks are analyzed under this model and code is generated. The generated code creates async executors for each priority level. An example of how tasks are scheduled in RTIC is shown in Figure 2.

This mechanism maps well to the ARM Cortex-M BASEPRI register [40], when available on the microcontroller. It is used to mask interrupts with a lower priority level than the one set in this register. If the BASEPRI register is not available, interrupt source masking is used instead.

**Embassy**: Embassy heavily relies on Rust’s asynchronous ecosystem. Tasks are asynchronous functions annotated with Embassy’s task macro. This macro implements Embassy traits necessary for these asynchronous functions to be used by the Embassy executor. Using this macro, the size of the function is known at compile time, meaning that no heap allocations are needed.

Embassy uses a cooperative scheduling model, where tasks need to yield control to allow other tasks to run. Yielding control is achieved by awaiting another Future, or by calling the yield_now function. Priority-based scheduling is also supported by Embassy. This is achieved by using different executors, where each executor has its own priority level. Executors are driven by interrupts. Using the hardware registers, interrupts can be prioritized. The interrupt with the highest priority level is used to drive the executor with the highest priority level. This way, the executor with the highest priority level is always scheduled first.

#### 4.2.2. Application Isolation and Memory Protection

**Tock**: Tock uses a combination of language-based and hardware-based isolation mechanisms. The language-based isolation is achieved by using capsules, which are untrusted by default and are, as such, not allowed to use unsafe code. Only capsules that implement core kernel functionalities are trusted and are allowed to use unsafe code.

The hardware-based isolation is implemented using the MPU of the microcontroller. The MPU is used to isolate the memory of each process. When a process tries to access memory outside its designated memory region, the MPU triggers a fault, preventing the operation.

**Hubris**: Hubris has its own build system, called xtask (not to be confused with the cargo-xtask tool). This system builds every task defined in the application configuration file as a separate binary. Each task defines how much Random Access Memory (RAM) and flash it needs. When all tasks are built, they are linked together into a single binary. Since each task is built as a separate binary, the tasks are isolated from each other. Furthermore, Hubris makes use of the MPU to isolate the memory of each task at runtime.

The disadvantage of this approach is that tasks that use the same library, e.g.,  a library for some specific communication protocol, are duplicated in the binary. This can lead to a larger binary size.

**RTIC and Embassy**: RTIC and Embassy do not use any form of application isolation or memory protection. Memory protection could be achieved by using the MPU of the microcontroller. The control registers of the MPU are often accessible using an HAL or PAC libraries.

#### 4.2.3. Inter-Process Communication (IPC)

**Tock**: Communication in Tock always occurs via the kernel using system calls. The system call triggers a context switch to the kernel, which then reads register values to determine how to handle the call. The context switch allows the kernel to switch to a privileged mode, allowing it to have full access to the system. There are seven system calls, split into two categories. The first category is for administrative purposes, containing three system calls: yielding control, requesting memory, and terminating the process. The second category is for interacting with capsules, containing four system calls. The *subscribe* command allows subscribing to an event, e.g., a specific interrupt. The *command* command allows invoking an action on a capsule. Finally, the *read-only allow* and *read-write allow* commands are for sharing data.

IPC between user applications is based on a client-server model. A process provides a service, implemented as a server, while other processes can share memory with the server to communicate. Sharing memory is conducted using the system calls. This shared memory is used to pass messages between the client and the server, or the other way around. The protocol of this communication is not specified by Tock, and is up to the developer to implement. The server and client can also communicate using the *subscribe* and *command* system calls.

Communication between user applications and capsules is achieved using the same system calls. Capsules often require dynamic memory, e.g., for storing metadata. Dynamic allocation is prohibited in the kernel, preventing a point of failure. Since capsules live in the kernel, they are not allowed to allocate memory. Instead, when capsules require dynamic memory, they can allocate memory from the process that invoked the capsule. This is implemented as *grants*, where the process provides a grant to the capsule using the *read-only allow* or *read-write allow* system calls.

IPC between capsules is not needed as they all live in the kernel space. Instead, they can directly call each other’s functions.

**Hubris**: IPC is based around *synchronous* messages, where sending a message blocks the sender, while it is waiting for a response. A receiver waiting for a message is also blocked until a message is received. Both tasks need to *rendezvous* in the right state. It is the kernel that copies the message from the sender to the receiver.

Since both tasks are blocked, one task is blocked while listening and the other while transmitting, no message queue is needed. A task can only continue its execution when the message is fully transmitted and received. This has the benefit that tasks cannot flood each other with messages. On the other hand, this can potentially lead to deadlocks. To solve this issue, Hubris only allows a task to send a message to a task with a higher priority.

Messages have a maximum size of 256 bytes. For bigger messages, the sender can use *leases*. A lease is a shared memory region that is shared between the sender and the receiver, where the sender leases the memory to the receiver. While the sender is waiting for a reply, the receiver has exclusive access to the memory. When the receiver is finished with the memory, it releases the lease, allowing the sender to access the memory again. This is an abstraction that can be expressed using the Rust borrowing rules, allowing the compiler to check if the memory is used correctly. This method also allows for higher priority tasks to send messages to lower priority tasks.

The protocol used for communication between tasks is defined by an experimental interface definition language called *Idol*. The language defines the messages that can be sent between tasks. At compile time, the message definitions are transpiled to Rust structs. These structs are then used to serialize and deserialize the messages. This is achieved using the popular serde library, a high-performance framework for serializing and deserializing Rust data structures.

**RTIC**: To share data between tasks, RTIC uses *channels*. A channel consists of multiple *senders* and one *receiver*. Tasks that want to send data use the sender construction, while the task that wants to receive data uses the receiver construction. The channel uses statically allocated memory to store the data.

The channels can be used asynchronously by software tasks. A sender transmitting data might need to wait for space in the channel to become available. Additionally, the receiver might need to wait for data to be available.

Hardware tasks can also use channels, but should not use the async features of the channel. These tasks should be short and should not block for a long time. A synchronous channel is, therefore, available for hardware tasks. Sending data over the channel might return an error when the channel is full.

**Embassy**: Embassy provides four different ways of IPC. The first way is by using *channels*. These channels can have multiple senders and multiple receivers. However, only one receiver can receive the message. The second way is with *priority* channels, where messages have priorities. Messages with higher priorities are received first. The third way is by means of *broadcast* channels, where multiple receivers can receive the same message. The last way is by using *signals*, which is the same as a channel, but with a signal message. When a new message is sent, the old message is overwritten.

#### 4.2.4. Hardware Interaction

**Tock**: Tock uses a Hardware Interface Layer (HIL) instead of the more commonly used embedded-hal, discussed in Section 3.1. This layer is hardware-agnostic and defines how components of the Tock OS interact with each other. Components can be hardware peripherals, such as SPI or I2C, but they can also be temperature sensors or buzzers. Hardware-specific Tock libraries implement the traits defined in the HIL. They also have board-specific libraries that initialize the microcontroller and board-specific Integrated Circuits (ICs) as well as the sensors or actuators.

**Hubris**: Interacting with hardware is achieved using driver abstractions, which are regular tasks that run in unprivileged mode, but can interact with the hardware. Hubris does not use the embedded-hal libraries for the hardware abstractions, as these libraries require privileged access. These drivers are usually split into two parts, platform-specific and platform-agnostic parts. However, this split is not well defined by Hubris. The platform-specific parts are written as driver crates and usually access the hardware. In some cases, these crates communicate using IPC with other driver crates, or a combination of both. The platform-agnostic parts are written as driver servers, which wrap the platform-specific crates and provide an IPC interface for other tasks.

**RTIC**: RTIC does not provide any hardware abstractions. The default way to interact with hardware is by using the HAL and PAC crates.

**Embassy**: Embassy provides its bespoke HALs, which are tailored to the asynchronous runtime of Embassy. Many hardware operations can be performed asynchronously, such as reading from a sensor or writing to a display. This makes the asynchronous runtime of Embassy a good fit for interacting with hardware. However, it is not necessary to use the HAL provided by Embassy, but then the hardware interaction will not be asynchronous. Embassy provides HALs for nRF microcontrollers from Nordic Semiconductor, STM32 microcontrollers from ST Microelectronics, and Raspberry Pi RP2040 microcontrollers.

#### 4.2.5. Networking

**Tock**: The network stack of Tock is written as a capsule. For the data link layer, an ieee802154 module provides the implementation of an IEEE 802.15.4 framer and a trait definition for an IEEE 802.15.4 device. The implementation of this trait is platform-specific and should provide buffers with the use of grants. Another capsule that is provided is the net capsule. This contains the implementation of IPv6 over 6LoWPAN, commonly used in IEEE 802.15.4-based networks. It supports compression for the Internet Control Message Protocol for IPv6 (ICMPv6) and UDP, with UDP the only supported transport layer protocol. Note that there is no support for Ethernet devices.

**Hubris and Embassy**: Hubris and Embassy use smoltcp as their network stack. For Hubris, the network stack is running in a dedicated task.

Embassy comes with a wrapper library, called embassy-net. It is a wrapper around the smoltcp library, running in its own Embassy task. At the time of writing, embassy-net only supports Ethernet over IPv4. Embassy does not contain an HAL implementation for devices using other physical media.

**RTIC**: RTIC does not provide a network stack. However, it is possible to use smoltcp with RTIC, as smoltcp is a standalone library.

### 4.3. Comparison Conclusion and Use Cases

Table 2 provides a summary of the features discussed in this section for Tock, Hubris, RTIC, and Embassy. For scheduling processes, we see that all of them provide a priority-based or round-robin-based scheduler. These are often relatively easy to implement compared to other scheduling mechanisms. Tock, however, provides more than one scheduling mechanism, such as purely cooperative-based scheduling or MLFQ scheduling.

In terms of Application Isolation and Memory Protection, both Tock and Hubris have mechanisms in place to isolate applications from each other. Tock and Hubris both use a combination of language-based and hardware-based isolation mechanisms, by using the MPU of the microcontroller. Neither RTIC nor Embassy provide any form of application isolation or memory protection.

Both frameworks can use the standard HALs of the Rust ecosystem. This enables the frameworks to run on a wider range of microcontrollers. Developers of HALs are not restricted to a specific OS or framework. However, this is not the case for Tock, as this OS already existed before the Rust ecosystem was mature and gained popularity early on.

For networking, Tock has its own network stack, whereas Hubris and Embassy use smoltcp. Note that Tock lacks support for Ethernet devices. Relying on smoltcp allows Hubris and Embassy to support a wider range of protocols. However, Tock’s network stack is more integrated with the OS.

In conclusion, real-time applications found in sensor networks, i.e., railway monitoring systems, disaster mitigation such as earthquake early warning systems, healthcare monitoring systems, and industrial automation systems, can benefit from the features provided by RTIC and Embassy. Both frameworks provide a lightweight runtime, suitable for real-time applications, with the latter providing a more complete library and network stack. For applications that require a more secure and full-fledged OS, Tock and Hubris are more suitable. Examples of such applications are automotive systems, threat monitoring systems, and privacy-sensitive applications.

## 5. Evaluation

In this section, we evaluate Tock, RTIC, and Embassy, discussed in Section 4. We do not evaluate Hubris, as it only supports a limited number of platforms, which are often developed by the Oxide Computer Company themselves. We evaluate this OS and these frameworks based on their scheduling latency and memory usage. The scheduling latency is crucial for real-time systems, where tasks need to be executed within a certain time frame. These measurements give an indication of how fast the OS can respond to external events. The memory usage is also important, as embedded devices often have limited resources.

Measurements are conducted on the nRF52840-DK development board from Nordic Semiconductor. It is based on a 32-bit ARM Cortex-M4 architecture, running at 64 MHz. For our measurements, we use the MSO2002B series mixed signal oscilloscope from Tektronix, OR, U.S., which has a resolution of 1 ns. We use the rustc v1.73.0-nightly-2023-07-30 compiler for all our examples, as it is the version required by Tock.

### 5.1. Interrupt Latency and Task Scheduling

For measuring the interrupt latency, we set a General Purpose Input/Output (GPIO) pin high at the beginning of the Interrupt Service Routine (ISR). This ISR is triggered by an input signal. The interrupt latency is the time between the input signal triggering the interrupt and the start of the ISR. We can measure this using the logic analyzer. The interrupt duration can be measured by setting the GPIO pin low at the end of the ISR. To measure the task scheduling latency, we use a task that toggles a GPIO pin. The time between the end of the ISR and the start of the task is measured using the logic analyzer. Adding up the interrupt latency, the interrupt duration and the scheduling latency gives the total latency. An overview of the measurement setup is shown in Figure 3.

The measurements are shown in Table 3. We perform the measurements for two different compiler optimization levels. The first half of the table shows the measurements when the software is compiled for speed optimization, while the second half shows the measurements when the software is compiled for size optimization. As input signal for the interrupt, we use a periodic signal such that multiple measurements can be conducted. This allows us to compute confidence intervals for the measurements, which are shown in the table. They are calculated using the normal distribution with a confidence level of 95%.

As expected, the total latency of Tock is highest. When the interrupt is triggered, a running task is preempted by the kernel, which then schedules the ISR. This ISR runs all functions that are registered to the interrupt. Furthermore, the OS deals with setting up the MPU correctly. This results in a higher interrupt latency, interrupt duration, and scheduling latency. Note that we conducted measurements for all four scheduling mechanisms in Tock. For each of them, the interrupt latency and interrupt duration are almost the same. The scheduling latency is different, with the highest scheduling latency for the MLFQ scheduler. This is expected, as the MLFQ scheduler has the most complex scheduling mechanism. The lowest scheduling latency is for the cooperative scheduler.

The total latency is lowest for RTIC, as they use the hardware to its full potential. We see that the total latency for Embassy is slightly higher than RTIC. Embassy has a slight overhead compared to RTIC, because of its asynchronous runtime.

For measurements in Table 3, only one task is scheduled, which is the task that toggles the GPIO pin. The scheduling latency in these measurements is, therefore, the smallest possible. Figure 4 shows the relation between the number of tasks and the total latency. For these measurements, additional tasks are running concurrently with the task that toggles the GPIO pin. We see that for RTIC, Embassy, and Tock with the priority scheduler and the MLFQ scheduler, the total latency increases linearly with the number of tasks. They have an R^2^ value of around 0.99 for a linear fit. For Tock with the round-robin scheduler and the cooperative scheduler, the total latency increases non-linearly.

### 5.2. Memory Requirements

The memory usage of an OS or framework is important, as embedded devices often have limited resources. We measure the memory usage of each OS or framework by looking at the binary size and the static memory usage. From the compiled binary, we extract the .text and .bss sections, which are the firmware size and the static memory usage, respectively. We use a simple application that blinks an LED for our measurement, as this is a very small application that shows the overhead of the OS or framework.

Table 4 shows the memory requirements of Tock, RTIC, and Embassy. The measurements show that RTIC has the smallest memory footprint, followed by Embassy. Embassy’s memory footprint is higher than RTIC, as it provides an executor for running asynchronous tasks. It is also known that asynchronous functions in Rust have a higher memory overhead compared to synchronous functions [41].

The memory footprint of Tock is split into two parts: the kernel and the application. The application part of Tock is very small, as it only contains system calls to the kernel. The kernel, however, is very large compared to RTIC and Embassy, as it contains many drivers and other functionalities. These drivers are potentially not used by applications, which is the case for our simple application. We tried to minimize the memory footprint of the Tock kernel by compiling without these drivers, which had a significant impact on the memory footprint.

## 6. Conclusions

In this work, we provide an introduction to embedded Rust and asynchronous programming in Rust. We discuss scheduling mechanisms, IPC, application isolation and memory protection, hardware interaction, and networking, and compare the design and implementation of these features in Tock, Hubris, RTIC, and Embassy. This comparison shows that each OS or framework has its own strengths and weaknesses. Tock is a full-fledged OS with support for networking and a wide range of microcontrollers. Hubris is a new OS that is still in development, but has a unique approach to application isolation. RTIC is a framework that is optimized for real-time systems, with a small memory footprint. Embassy is a framework that is optimized for asynchronous programming, with a slightly higher memory footprint compared to RTIC, but with a more flexible, asynchronous-capable HAL. We also provide a performance evaluation of Tock, RTIC, and Embassy, based on their scheduling latency and memory usage. The results show that RTIC has the lowest scheduling latency and memory footprint, followed by Embassy. Tock has the highest scheduling latency and memory footprint, as it is a full-fledged OS.

As future work, these Rust-based OSs and frameworks can be further evaluated on other aspects, such as energy consumption, scalability, or performance with more complex applications. They could also be compared to other OSs and frameworks, such as RIOT-OS, Zephyr, and Contiki-NG, written in other programming languages. Further studies could also explore their security features, ease of integration with hardware, and long-term maintenance requirements.

## Figures and Tables

**Figure 1 sensors-24-05818-f001:**
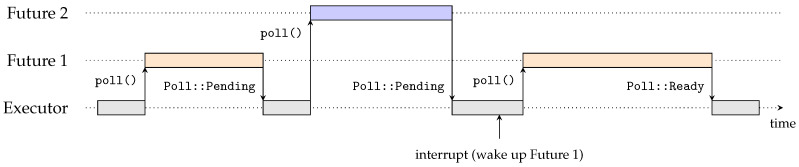
Asynchronous model in Rust with an executor and 2 futures. The executor polls the futures, which are executed until they are ready. A future returns Poll::Pending when it needs to wait for a resource, and Poll::Ready when it is ready. The executor is notified when a future is ready to be polled again.

**Figure 2 sensors-24-05818-f002:**
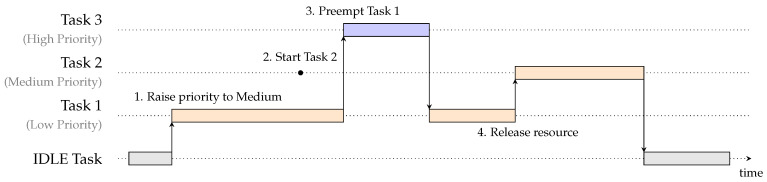
Example of Stack Resource Policy-based scheduling in RTIC with 3 tasks, with Task 1 having the lowest priority and Task 3 the highest. Task 1 and Task 2 share the same resource. When Task 1 is scheduled, it raises the running priority to medium, which is the highest priority of the resources it is using (shown in step 1). Task 2 is scheduled, but cannot preempt Task 1 as it requires the same resource (shown in step 2), preventing a deadlock where Task 2 would wait indefinitely for the occupied resource. Task 3 is scheduled, preempting Task 1 (shown in step 3) as it has a higher priority. When Task 3 is finished, the execution of Task 1 is resumed. Task 1 releases the resource and lowers the priority to its original level (shown in step 4). Task 2 can now preempt Task 1.

**Figure 3 sensors-24-05818-f003:**
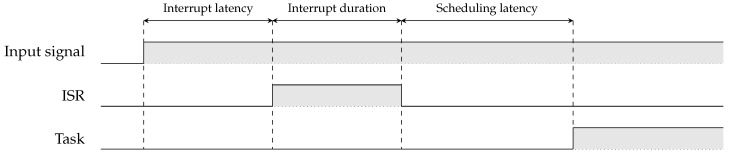
*Interrupt latency* and *scheduling latency* measurement setup. The interrupt latency is the time between the interrupt being triggered and the start of the ISR. The interrupt duration is the time it takes to handle the ISR. The scheduling latency is the time it takes to schedule a task. The microcontroller should be in an idle state such that the interrupt does not preempt a running task. Otherwise, the scheduling latency is not measured correctly, as after the interrupt handling, another task would resume its execution before task scheduling occurs.

**Figure 4 sensors-24-05818-f004:**
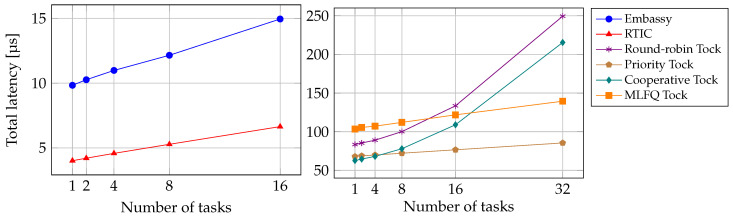
Total latency measurements for different number of tasks for Embassy, RTIC, and Tock. For Tock with the round-robin and cooperative scheduler, the total latency increases non-linearly with the number of tasks. All other schedulers show a linear increase in total latency.

**Table 1 sensors-24-05818-t001:** General overview of Operating Systems and frameworks in Rust for embedded systems.

	License	Targeted Platforms	Contr. ^1^	Git Commits	kLOC	GitHub Stars
Tock (2.1.1) Operating System	Apache 2.0 or MIT	ARM Cortex-M RISC-V ^2^	186	13,803	193.5	5271
Hubris (1.0.0) Operating System	Mozilla Public License 2.0	ARM Cortex-M	49	1840	89.2	2908
RTIC (2.0.0) Framework	Apache 2.0 or MIT	ARM Cortex-M	83	1707	13.7	1708
Embassy (0.1.0) Framework	Apache 2.0 or MIT	ARM Cortex-M RISC-V ^2^	353	8393	134.3	4925

^1^ Number of active contributors. ^2^ Experimental support.

**Table 2 sensors-24-05818-t002:** Summary of discussed key aspects for Tock, Hubris, RTIC, and Embassy.

	Scheduling	Inter-Process Communication (IPC)	App. Isolation	Hardware Interaction	Network Stack
Tock OS	One of: • Round-robin • Cooperative • Priority-based • MLFQ	• System calls • Shared memory	MPU	Bespoke HAL	Bespoke stack: • IEEE 802.15.4 framer • 6LoWPAN • ICMPv6 • UDP
Hubris OS	Priority-based and cooperative	Synchronous message-passing	MPU	Bespoke HAL	smoltcp based: • Ethernet framer • IEEE 802.15.4 framer • IPv4/IPv6 • 6LoWPAN • ICMPv4/ ICMPv6 • TCP/UDP
RTIC Framework	• Priority-based and preemptive • Hardware-accelerated	Channel based	-	embedded-hal	-
Embassy Framework	Priority-based and cooperative	Channel based	-	• Bespoke HAL (async) • embedded-hal	smoltcp based: • Ethernet framer • IEEE 802.15.4 framer • IPv4/IPv6 • 6LoWPAN • ICMPv4/ ICMPv6 • TCP/UDP

**Table 3 sensors-24-05818-t003:** Interrupt latency, interrupt duration, and scheduling latency measurements. The measurements are conducted on the nRF52840-DK development board. The software is compiled once for speed optimization (opt-level = 3) and once for size optimization (opt-level = "z"). Confidence intervals are calculated using the normal distribution with a confidence level of 95%.

Optimized for Speed	Interrupt Latency (µs)	Interrupt Duration (µs)	Scheduling Latency (µs)	Total Latency (µs)
*Embassy*	0.822 (0.003)	4.844 (0.003)	4.164 (0.001)	9.830 (0.003)
RTIC	0.668 (0.003)	2.207 (0.001)	1.135 (0.003)	4.010 (0.004)
round-robin Tock	15.683 (0.005)	15.683 (0.002)	52.058 (0.003)	83.424 (0.005)
priority Tock	15.741 (0.006)	15.635 (0.004)	36.510 (0.003)	67.886 (0.005)
cooperative Tock	15.737 (0.005)	15.634 (0.004)	31.323 (0.003)	62.694 (0.005)
MLFQ Tock	15.596 (0.005)	15.652 (0.002)	72.166 (0.003)	103.414 (0.004)
**Optimized for Size**	**Interrupt Latency (µs)**	**Interrupt Duration (µs)**	**Scheduling Latency (µs)**	**Total Latency (µs)**
Embassy	1.03 (0.01)	6.573 (0.002)	5.7 (0.4)	13.3 (0.4)
RTIC	0.995 (0.003)	2.825 (0.002)	1.432 (0.005)	5.252 (0.006)
round-robin Tock	15.237 (0.005)	29.917 (0.007)	74.656 (0.003)	119.810 (0.006)
priority Tock	15.232 (0.006)	29.48 (0.02)	56.279 (0.004)	100.99 (0.02)
cooperative Tock	15.375 (0.006)	30.12 (0.02)	50.454 (0.004)	95.95 (0.01)
MLFQ Tock	15.422 (0.006)	28.871 (0.008)	84.33 (0.07)	128.63 (0.08)

**Table 4 sensors-24-05818-t004:** Memory requirements for an application blinking an LED every 250 milliseconds. The application is compiled for the nRF52840-DK development board, and optimized for size (opt-level = "z").

	Text (byte)	Data (byte)	BSS (byte)	Total (byte)
Embassy	9588	112	34,264	43,964
RTIC	6864	0	40	6904
Kernel Tock	200,708	36	34,492	235,236
Simple kernel Tock	40,964	32	10,960	51,956
Application Tock	286	0	0	286

## Data Availability

The data that support the findings of this study are available from the corresponding author, T.V., upon reasonable request.

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
