# Peer review of "Overview of Embedded Rust Operating Systems and Frameworks"

_sensors, 2024, doi:10.3390/s24175818_

Round 1

Reviewer 1 Report

Comments and Suggestions for Authors

The paper aims to compare Rust-based OS for sensor nodes. The authors explain the benefits of using Rust over traditional C++ based OS and give details on memory usage and related security issues. 

They choose 4 OS for the comparism and provide theoretical comparison of features and measurements of interrupt timing to measure performance.

The paper is well structured and easy to follow.

However the following points should be improved prior to publication:

- the presented  comparison focusses on Rust-based OS only and does not cover features available in traditional OS. Therefore it is hard to assess whether they are suitable as replacement or which features are missing to enable this.

- Please also discuss in more detail the pros and cons of Rust versus C++ based OS and give a recommendation which studied variant would be most suitable for example usecases.

- Please highlight some insightss on future development and how you plan to extend the study.

Comments on the Quality of English Language

The language is good and easy to follow. Please carefully read and check for minor grammar and spelling mistakes.

Author Response

The authors would like to thank the reviewer for their constructive and insightful feedback. We have tried to address all comments and suggestions. Below, we describe each point into more detail.

Point 1: the presented comparison focusses on Rust-based OSs only and does not cover features available in traditional OSs. Therefore, it is hard to assess whether they are suitable as a replacement or which features are missing to enable this.

Indeed, we focus on Rust-based OSs only. In our overview, we compare the same features as discussed in the work of Javed et al. [17]. We think that these features are the ones found in traditional OSs for sensor nodes. We apologize if this was not clear. Lines 181 – 183 were added to clarify this. Lines 56 – 64 from the related work section are also referring to features available in OSs, which we believe are the ones found in traditional OSs.

Lines 181 – 183: “The following sections go deeper into the core features of an OS: task scheduling, application isolation and memory protection, IPC, hardware interaction, and networking.”

Point 2: please discuss in more detail the pros and cons of Rust versus C++ based OSs and give a recommendation which studied variant would be most suitable for example usescases.

As highlighted in the abstract, our research paper focusses on Rust-based OSs. We understand the importance of comparisons with OSs written in other languages, such as C or C++, but our current scope is limited to Rust-based OSs. We added lines 570577 to provide use cases that best match each of the studied operating systems and frameworks.

Lines 570 – 577: “In conclusion, real-time applications found in sensor networks, i.e., railway monitoring systems, disaster mitigation such as earthquake early warning systems, healthcare monitoring systems, and industrial automation systems, can benefit from the features provided by RTIC and Embassy. Both frameworks provide a lightweight runtime, suitable for real-time applications, with the latter providing a more complete library and network stack. For applications which require a more secure and full-fledged OS, Tock and Hubris are more suitable. Examples of such applications are automotive systems, threat monitoring systems, and privacy-sensitive applications.”

Point 3: please highlight some insights on future development and how you plan to extend the study.

We added the following lines about future development to Section 6 Conclusion:

Lines 662 – 667: “As future work, these Rust-based OSs and frameworks can be further evaluated on other aspects, such as energy consumption, scalability, or performance with more complex applications. They could also be compared to other OSs and frameworks, such as RIOT-OS, Zephyr and Contiki-NG, written in other programming languages. Further studies could also explore their security features, ease of integration with hardware, and long-term maintenance requirements.”

Reviewer 2 Report

Comments and Suggestions for Authors

The paper showed a comparison of RUST based OS that are available. It presents a good summary  and comparison

Author Response

The authors would like to thank the reviewer for their feedback.

Reviewer 3 Report

Comments and Suggestions for Authors

The paper presents an overview and comparison of embedded operating systems and frameworks in the Rust programming language. Advantages of the Rust language compared to C are briefly presented in the introduction, followed by an overview of the related papers on embedded OS. An embedded Rust ecosystem and its features are described. The paper presents detailed overview of two Rust-based OS and two frameworks and evaluation of selected systems on ARM Cortex-M4 platform.

The paper is largely based on overview of the important embedded OS features. The actual evaluation section is very short and results are obtained with a very basic experiments, so the title could be misleading. Consider changing Evaluation to Overview in the title unless you considerably improve the evaluation section.

Are the measurements from the Table 3 repeatable? When measuring interrupt latency, you use input button, which can have debouncing issues. Would it be better to use periodic signal generator as an input?
How is the scheduling latency dependent on the number of tasks?

In 4.1.3 you state that RTIC has experimental support for RISC V, which is contradictory to the data in Table 1.

Author Response

The authors would like to thank the reviewer for their constructive and insightful feedback. We have tried to address all comments and suggestions. Below, we describe each point into more detail.

Point 1: The actual evaluation section is very short and results are obtained with very basic experiments, so the title could be misleading. Consider changing Evaluation to Overview in the title unless you considerably improve the evaluation section.

Indeed, the evaluation section is short and basic, and we agree to change the title.

Point 2: When measuring interrupt latency, you use an input button, which can have debouncing issues. Would it be better to use a periodic signal generator as an input?

We updated our measurements by using a periodic signal as an input. We have multiple measurements for the whole sequence (interrupt latency, interrupt duration and scheduling latency), which allowed us to compute confidence intervals. We added them in the table. Section 5.1 was updated to better explain the measurements.

Point 3: How is the scheduling latency dependent on the number of tasks?

This is indeed an interesting and valuable point. To address it, we've added measurements that demonstrate the relationship between the number of tasks and scheduling latency.

We added Figure 4 showing the relationship between the number of tasks and the scheduling latency. We only did the measurements with the optimized for speed compiler settings. Lines 620 – 627 were added to explain the new findings.

Lines 620 – 627: “For measurements in Table 3, only one task is scheduled, which is the task that toggles the GPIO pin. The scheduling latency in these measurements is therefore the smallest possible. Figure 4 shows the relation between the number of tasks and the total latency. For these measurements, additional tasks are running concurrently with the task that toggles the GPIO pin. We see that for RTIC, Embassy, and Tock with the priority scheduler and the MLFQ scheduler, the total latency increases linearly with the number of tasks. They have an R2 value of around 0.99 for a linear fit. For Tock with the round-robin scheduler and the cooperative scheduler, the total latency increases non-linearly.”

Point 4: In 4.1.3 you state that RTIC has experimental support for RISC-V, which is contradictory to the data in Table 1.

Thank you for pointing this out. The support for RISC-V is indeed experimental, and we now clarify this in the table by adding a superscript 2 to indicate its experimental status.

Reviewer 4 Report

Comments and Suggestions for Authors

- This Manuscript is neither a review nor a research paper.

- Authors present details about a programming tool.

- What is the main aim and their contribution?

- What are the designated applications for the readers?

- Some application examples can be added with comments.

Author Response

The authors would like to thank the reviewer for their feedback.

Point 1: this manuscript is neither a review nor a research paper.

Our manuscript aims to serve as an exploration of Rust-based operating systems and frameworks, offering an analysis of some of their most important features and performance paramters. While it does not follow the traditional style of a review or research paper, we believe it can provide valuable insights for researchers interested in Rust-based embedded operating systems. This is already pointed out in the abstract and introduction.

Point 2: Authors present details about a programming tool.

Our focus was on providing an overview and basic performance comparison of Rust-based operating systems and frameworks, which we see as more than just programming tools. They represent a new approach to system development with unique advantages, such as memory safety. The advantages are underlined in Section 1 Introduction and Section 3 Preliminaries.

Point 3: what is the main aim and the contributions?

The main aim of our paper is to explore the capabilities, strengths, and potential applications of Rust-based operating systems and frameworks. Our key contributions include a detailed analysis of these systems, performance evaluations, and insights into their suitability for various applications. We improved our performance evaluation by using a periodic signal as an input instead of the button. Additionally, we now have multiple measurements for the whole sequence (interrupt latency, interrupt duration and scheduling latency), which allowed us to compute confidence intervals. We added them in the Table 3. Section 5.1 was updated to better explain the measurements. Furthermore, we added Figure 4 showing the relationship between the number of tasks and the scheduling latency. Lines 620 – 627 were added to explain the new findings. We added lines 570577 to provide use cases that best match each of the studied operating systems and frameworks.

Lines 620 – 627: “For measurements in Table 3, only one task is scheduled, which is the task that toggles the GPIO pin. The scheduling latency in these measurements is therefore the smallest possible. Figure 4 shows the relation between the number of tasks and the total latency. For these measurements, additional tasks are running concurrently with the task that toggles the GPIO pin. We see that for RTIC, Embassy, and Tock with the priority scheduler and the MLFQ scheduler, the total latency increases linearly with the number of tasks. They have an R2 value of around 0.99 for a linear fit. For Tock with the round-robin scheduler and the cooperative scheduler, the total latency increases non-linearly.”

Lines 570 – 577: “In conclusion, real-time applications found in sensor networks, i.e., railway monitoring systems, disaster mitigation such as earthquake early warning systems, healthcare monitoring systems, and industrial automation systems, can benefit from the features provided by RTIC and Embassy. Both frameworks provide a lightweight runtime, suitable for real-time applications, with the latter providing a more complete library and network stack. For applications which require a more secure and full-fledged OS, Tock and Hubris are more suitable. Examples of such applications are automotive systems, threat monitoring systems, and privacy-sensitive applications.”

Point 4: what are the designated applications for the reader? Some application examples can be added with comments.

We added lines 570577 to clarify possible applications.

Lines 570 – 577: “In conclusion, real-time applications found in sensor networks, i.e. railway monitoring systems, disaster mitigation such as earthquake early warning systems, healthcare monitoring systems, and industrial automation systems, can benefit from the features provided by RTIC and Embassy. Both frameworks provide a lightweight runtime, suitable for real-time applications, with the latter providing a more complete library and network stack. For applications which require a more secure and full-fledged OS, Tock and Hubris are more suitable. Examples of such applications are automotive systems, threat monitoring systems, and privacy-sensitive applications.”

Round 2

Reviewer 4 Report

Comments and Suggestions for Authors

- The Authors tried to answer the comments.

- It is better to select a critical problem and solve it briefly using this tool development. 

- Add certain suitable algorithms and mathematics.

Author Response

Comment 1:  It is better to select a critical problem and solve it briefly using this tool development.

The "tools" that we evaluated are Rust-based Operating Systems. The problems OSs solve are: task scheduling, application isolation and memory protection, IPC, hardware interaction, and networking, for which we give an overview and performance study.

Comment 2: Add certain suitable algorithms and mathematics.

The paper focuses on an empirical analysis with performance measurements and analysis of results.  We did not mathematically model these OSs.

We hope we understood your comments such that our answers are a response to your remarks.